# Reduced running performance and greater perceived exertion, but similar post-exercise neuromuscular fatigue in tropical natives subjected to a 10 km self-paced run in a hot compared to a temperate environment

**Jefferson F. C. Rodrigues, Júnior** [1], **Thiago T. Mendes**[2], **Patrícia F. Gomes**[1], **Emerson Silami-Garcia**[1,3], **Fabiano T. Amorim**[4], **Mário N. O. Sevilio, Jr.**[5], **Fabrício E. Rossi**[6,7], **Samuel P. Wanner** [1] *

1 Exercise Physiology Laboratory, School of Physical Education, Physiotherapy and Occupational Therapy, Universidade Federal de Minas Gerais, Belo Horizonte, Minas Gerais, Brazil, 2 Department of Physical Education, Faculty of Education, Universidade Federal da Bahia, Salvador, Bahia, Brazil, 3 Department of Sports, School of Physical Education, Physiotherapy and Occupational Therapy, Universidade Federal de Minas Gerais, Belo Horizonte, Minas Gerais, Brazil, 4 Department of Health, Exercise and Sports Sciences, University of New Mexico, Albuquerque, New Mexico, United States of America, 5 Health Sciences Center, Universidade Federal do Maranhão, São Luís, Maranhão, Brazil, 6 Immunometabolism of Skeletal Muscle and Exercise Research Group and Laboratory of Muscle Performance, Department of Physical Education, Universidade Federal do Piauí, Teresina, Piauí, Brazil, 7 Graduate Program in Science and Health, Universidade Federal do Piauí, Teresina, Piauí, Brazil

* samuelwanner@ufmg.br

## Abstract

Environmental heat stress impairs endurance performance by enhancing exercise-induced physiological and perceptual responses. However, the time course of these responses during self-paced running, particularly when comparing hot and temperate conditions, still needs further clarification. Moreover, monitoring fatigue induced by exercise is paramount to prescribing training and recovery adequately, but investigations on the effects of a hot environment on post-exercise neuromuscular fatigue are scarce. This study compared the time course of physiological and perceptual responses during a 10 km self-paced treadmill run (as fast as possible) between temperate (25°C) and hot (35°C) conditions. We also investigated the changes in countermovement jump (CMJ) performance following exercise in these two ambient temperatures. Thirteen recreational long-distance runners (11 men and 2 women), inhabitants of a tropical region, completed the two experimental trials in a randomized order. Compared to 25°C, participants had transiently higher body core temperature ($T_{CORE}$) and consistently greater perceived exertion while running at 35°C ($p < 0.05$). These changes were associated with a slower pace, evidenced by an additional 14 ± 5 min (mean ± SD) to complete the 10 km at 35°C than at 25°C ($p < 0.05$). Before, immediately after, and 1 h after the self-paced run, the participants performed CMJs to evaluate lower limb neuromuscular fatigue. CMJ height was reduced by 7.0% (2.3 ± 2.4 cm) at 1 h after the race ($p < 0.05$) compared to pre-exercise values; environmental conditions did not influence this reduction. In conclusion, despite the reduced endurance performance, higher perceived

**Data Availability Statement:** All relevant data are within the paper and its Supporting information files.

**Funding:** This study was financed by the Conselho Nacional de Desenvolvimento Científico e Tecnologico (CNPq/Brazil; www.gov.br/cnpq/; grant number: 427095/2018-2). This study was also supported by CNPq/MCTIC/CAPES/FNDCT/PROANTAR (grant number 442645/2018-0). In addition, SPW receives a fellowship from the CNPq for being a productive researcher (grant number: 315199/2021-0). The funders had no role in study design, data collection and analysis, decision to publish, or preparation of the manuscript.

**Competing interests:** We have read the journal's policy and the authors of this manuscript have the following competing interests: Samuel Penna Wanner, who is the corresponding author of this manuscript, currently works as an Academic Editor for PLOS One. The other authors have declared that no competing interests exist.

exertion, and transiently augmented $T_{CORE}$ caused by environmental heat stress, post-exercise neuromuscular fatigue is similar between temperate and hot conditions. This finding suggests that the higher external load (faster speed) at 25°C compensates for the effects of more significant perceptual responses at 35°C in inducing neuromuscular fatigue.

## Introduction

Robust evidence indicates that environmental conditions influence fatigue perception and performance during prolonged physical exercises [1–4]. For example, high ambient temperature ($T_A$) conditions impair endurance performance [5, 6], and this effect results from increased body core hyperthermia, leading to changes in brain electrical activity, augmented perceived exertion, and reduced neuromuscular drive [7, 8]. The accelerated fatigue during exercise-heat stress is also explained by enhanced cardiovascular strain [9] and altered thermal perception [10]. Even though the role of physiological and perceptual responses in impairing endurance performance under environmental heat stress is well acknowledged in the literature, the time course of these responses, particularly when comparing a self-paced treadmill run between warm/hot and cool/temperate conditions, was less studied.

While some studies only reported physiological and perceptual responses at the end of self-paced exercises at different $T_As$ [11–13], Marino et al. [14, 15] described the time course of these responses in athletes subjected to 8 km treadmill runs in hot (35°C) and cool (15°C) conditions. However, in the experiments conducted by Marino and colleagues, the athletes ran at 70% of their peak speed for 30 min at 15°C or 35°C before the 8 km runs; therefore, athletes already started the self-paced exercise at 35°C with higher body core temperature ($T_{CORE}$), heart rate (HR), and rating of perceived exertion (RPE) than at 15°C. Also important, whether the changes caused by exercise-heat stress are evident in heat-acclimatized individuals still requires further investigation since inhabitants of tropical climates have adaptations that improve heat tolerance and reduce physiological and perceptual strain when exposed to hot conditions [16]. Thus, the current study fills these literature gaps by describing the time course of physiological and perceptual responses in recreational athletes, inhabitants of a tropical region, subjected to prolonged self-paced exercises in temperate and hot conditions.

Aside from changes occurring during exercise, another point that merits investigation is the effect of a self-paced run under hot environmental conditions on post-exercise neuromuscular performance and, consequently, neuromuscular fatigue. Monitoring muscle fatigue is paramount to adequately prescribing training loads and recovery [17, 18]. Fatigue is defined as an exercise-induced reduction in maximal voluntary muscle force or power production [19] and can be assessed by measuring physiological and perceptual variables and physical performance [20–22]. Due to its validity, reliability, and low cost, performance in the countermovement jump (CMJ) test has been widely used to monitor athletes' lower limb neuromuscular fatigue during and after training sessions [23–25]. For example, weekly averages of CMJ height significantly correlated with perceived exertion, distance covered, and training zone in elite middle- and long-distance runners [26]. Furthermore, from an acute fatigue perspective, CMJ height was reduced at 24 h and 48 h post-match compared to 24 h before the match in young Turkish football players [27].

Under hot conditions, a fixed-intensity endurance exercise to fatigue induces a greater reduction in neuromuscular performance compared to temperate conditions [8, 28]. However, exercising at a fixed intensity (*i.e.*, open-loop exercise) does not reproduce the demands of real-world sports competitions [29]. Therefore, we decided to investigate the influence of a hot environment on the level of neuromuscular fatigue induced by a self-paced (*i.e.*, closed-loop)

endurance exercise, in which the individuals adjust running speed to choose the best strategy to complete a predetermined distance as fast as possible [29–31]. In this regard, Périard et al. [32] reported that the loss of force production following self-paced cycling exercise in the heat was not exacerbated compared to the loss observed under temperate conditions. These authors investigated force production and voluntary activation of the knee extensors while subjects performed a maximal voluntary isometric contraction. In the present study, we propose measuring performance in a dynamic exercise, commonly used in training settings to monitor fatigue, to understand further whether the environmental heat stress interferes with the level of post-exercise neuromuscular fatigue.

Therefore, the current study compared the time course of physiological and perceptual responses in tropical natives during a 10 km self-paced treadmill run between temperate (25°C) and hot (35°C) conditions. In addition, this study investigated the changes in counter-movement jump (CMJ) performance following 10 km runs in $T_A$s mentioned earlier. Considering the previous evidence (*e.g.*, [32]), we expected that running 10 km in the heat, compared to temperate conditions, would increase RPE and $T_{CORE}$ at exercise completion (end-$T_{CORE}$) but would not exacerbate lower limb neuromuscular fatigue and the exercise-induced HR increase. Because our participants were heat acclimatized (*i.e.*, they lived in a tropical climate in Brazilian's northeast region), we also hypothesized that some effects induced by exercise-heat stress would be less evident in the current than in the previously-published studies.

## Methods

### Participants and ethical care

Thirteen recreational long-distance runners (11 men and 2 women) were recruited and completed the two experimental trials. All runners were born in Teresina (Brazil), a city located in a tropical region (latitude: 5°05'20" S, longitude: 42°48'07" W), with a monthly (calculated from 01/2013 to 12/2022; mean ± SD) average $T_A$ and relative humidity of 28.0 ± 1.3°C and 72 ± 10%, respectively [33]. Of note, the maximum $T_A$ surpasses 31°C in all months of the year and corresponds to a monthly average of 34.6 ± 2.2°C [33]. In addition, at the time of the experiments, the 13 participants lived in Teresina and were therefore considered heat-acclimatized individuals.

The inclusion criteria were: 1) to train regularly for at least two years; 2) to have a weekly training frequency of at least five days; 3) the absence of lower limb injuries in the six months before the experiments; 4) to not use any medication and nutritional supplements during the experiments; 5) to have completed a 10 km outdoor run in less than 45 min (men) or 55 min (women) in the six months before the experiments. This required performance could have been attained during training sessions or competitions. Of note, the two women participants used monophasic oral contraceptives and were tested during the active phase of the contraceptive pill, from the 2nd to the 21st day of use [34].

The study was approved by the Research Ethics Committee of the Universidade Federal do Maranhão (protocol number 1.548.709). The procedures conformed to the 1964 Helsinki Declaration and its later amendments. Moreover, the participants signed an informed consent form upon agreeing to participate in this study.

### Sample size calculation

Sample size calculation was performed a priori using data from pilot experiments (n = 5) investigating the effect of a 10 km run under temperate conditions on the CMJ height. These experiments indicated that CMJ would possibly reduce at 1 h post-running, with an effect size (*i.e.*, Cohen's d) of 0.87. We then used the GPower software (version 3.1.9.7, Universität

Düsseldorf, Germany) to calculate the required sample size according to the following additional parameters: the difference between two dependent means (matched pairs) when using a t-test, an alpha error = 0.05, and a power = 0.90. This calculation indicated that 13 participants were needed to identify significant changes in the CMJ height 1 h after finishing the 10 km.

## Experimental design

The current experiments followed a crossover design, with each participant subjected to familiarization procedures and two experimental trials. Therefore, the participants visited the laboratory three times.

During the first visit, the experimental procedures were explained to the volunteers, who had the opportunity to ask questions about the protocol. Anthropometric measurements were performed, and then the participants were subjected, in the following order, to a familiarization session with CMJ, a maximal incremental cardiopulmonary exercise test, and a familiarization session with running on a motorized treadmill.

Experimental trials were carried out during the second and third visits (Fig 1). The volunteers were subjected to a 10 km self-paced run in a temperate (25˚C) or hot environment (35˚C), with relative humidity controlled between 40 and 50%. The order of the experimental trials was randomized. The interval between trials varied between 72 h and 96 h, and they were conducted at the same time of day (less than 1-hour difference when exercises were initiated), always in the afternoon (between 2:30 p.m. and 6:00 p.m.), to prevent circadian rhythm from interfering with the interpretation of our findings.

The following variables were measured during exercise at 2 km intervals: HR, $T_{CORE}$, RPE, and thermal sensation. CMJ height, an indicator of lower limb neuromuscular fatigue, was measured before, immediately after, and 1 h after the 10 km run.

## Procedures

**Anthropometric measurements.** Stature and body mass were determined, respectively, using a measuring tape and a bioimpedance scale (precision: 100 g; model HBF 514c, Omron Corporation, Kyoto, Japan). Skinfold thickness was measured in triplicates by the same

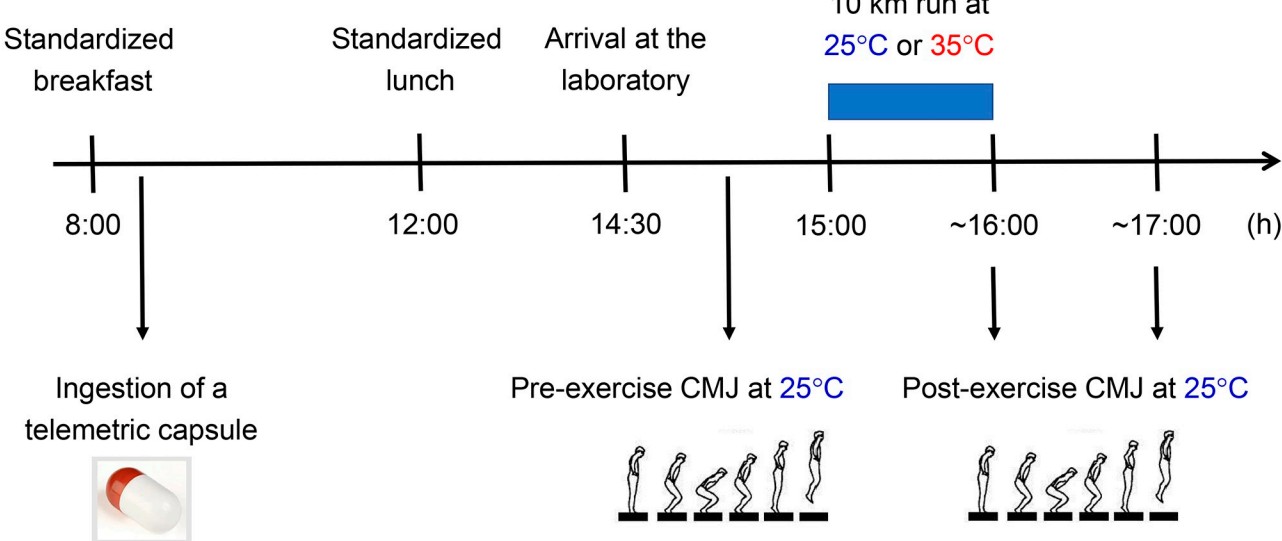

**Fig 1. Timeline of the experimental trials.** CMJ = countermovement jump.

investigator at seven sites–triceps, subscapular, pectoral, mid-axilla, abdominal, supra iliac, and mid-thigh–using a skinfold caliper (Lange, MI, USA). The body fat percentage was calculated according to Jackson and Pollock's equations [35]. Body surface area was also calculated according to the equation proposed by Du Bois & Du Bois [36].

**Familiarization with CMJ.**   Initially, the individuals were subjected to a 10 min warm-up on a treadmill at a speed corresponding to 70% (*i.e.*, 8.4 ± 1.0 km·h$^{-1}$) of the average speed attained during their best 10 km performance. Previous evidence indicates that warming up muscles increases short-term high-intensity physical performance [37, 38]. This warm-up also preceded the CMJs performed before and 1 h after the 10 km self-paced exercise during the second and third visits.

After the standardized warm-up, the volunteers were familiarized with the jumping technique by performing 3 sets of 3 CMJ repetitions, with a 30-second interval between sets [39]. The participants initiated the jump from the upright position, with parallel feet on a force platform and hands on the hips to neutralize the influence of the upper limbs. At the examiner's signal, the individuals performed a continuous movement, with the concentric action preceded by a preparatory movement corresponding to an eccentric action until approximately 90˚ of knee flexion [39].

**Cardiopulmonary exercise testing.**   After being familiarized with CMJs, the participants were subjected to an incremental cardiopulmonary exercise test to estimate peak oxygen consumption (VO$_{2peak}$) using a modified Åstrand protocol [40]. The test started at a speed of 11.3 km·h$^{-1}$ and an incline of 0%. Speed was maintained at 11.3 km·h$^{-1}$ throughout the protocol, with the incline increasing by 2.5% every 2 min; the speed and increment in incline were selected to exhaust most participants within 7 to 10 min of the test [41]. Resting VO$_2$ was summed to the VO$_2$ corresponding to both the horizontal and vertical components of the last stage completed to estimate VO$_{2peak}$. As proposed by the American College of Sports Medicine [42], the following criteria were used to interrupt the test: 1) the participant requested to interrupt the exercise or scored 10 on Borg's RPE scale [43]; 2) the presence of malaise signs, such as a pale appearance to the skin or cyanosis (signs of poor perfusion); 3) unusual shortness of breath; or 4) failure of the equipment. A chest-strapped monitor (Polar H10, Kempele, Finland) was used for the continuous HR measurement.

**Familiarization with the treadmill.**   Forty min after the cardiopulmonary exercise test, participants were further familiarized with the motorized treadmill (model Evoque Jet 6, TGR Fitness, Blumenau–SC, Brazil) by running 5 km at the same speed as the warm-up described earlier (*i.e.*, 8.4 ± 1.0 km·h$^{-1}$). The incremental test and the familiarization session were performed under temperate conditions (T$_A$ = 25.5 ± 0.4˚C and relative humidity = 51.9 ± 2.1%).

**Experimental trials.**   On the day of the experiments, around 8:00 a.m., the volunteers ingested a telemetric capsule to allow the measurement of gastrointestinal temperature (*i.e.*, an index of T$_{CORE}$). At noon, the participants had lunch and arrived at the laboratory at 2:30 p. m., where they stayed in a temperate environment (~25˚C) until the pre-race variables, body mass, and hydration status were measured. We ensured that the participants were not dehydrated when they started running, as indicated by a urine specific gravity equal to or less than 1.029 [44].

The participants were instructed to complete the predetermined distance as fast as possible and, while running, had only visual feedback of the distance covered. The environmental conditions selected for the experiments were based on a previous study from our laboratory, which showed that the time to complete a 30 km time trial on a cycle ergometer was 9% longer at 35˚C than at 24˚C [4]. T$_A$ inside the experimental room was warmed up with the help of an electric heater (model AB1100N, Britânia, Brazil) or cooled down with a split air conditioning system (Springer Midea, Brazil). Whenever required, this split was also used in the

dehumidification function. The dry and wet $T_A$s were measured using the Thermal Stress Meter (TGD 400, Instrutherm, Brazil) and were used to calculate the relative humidity.

During the study period, the participants were instructed by a nutritionist to eat a standardized diet containing 6.5 g/kg of carbohydrates, 2.0 g/kg of fat, and 1.5 g/kg of protein, over five daily meals (breakfast, lunch, snack, dinner, and supper), according to the recommendations of the Academy of Nutrition and Dietetics, the Dietitians of Canada, and the American College of Sports Medicine [45]. The diets were prescribed individually, according to the participant's body mass. Food was weighed by the participants with an electronic kitchen scale (precision: 1.0 g) to ensure standardization.

## Variables measured

**Physical performance.** The endurance performance corresponded to the time elapsed between the beginning and end of the 10 km self-paced run. The average speed at 2 km intervals was also recorded to understand how participants selected exercise intensity during the race. The time elapsed was measured with a stopwatch (precision of 0.01 s), and average speed was calculated as the distance traveled divided by a given time interval.

Lower limb neuromuscular power was determined by measuring the CMJ height. Athletes performed the CMJs on a custom-made force platform (Inovação em Tecnologia Esportiva, Belo Horizonte–MG, Brazil) with a 0.1 cm precision. The following equation accounted for the jump height: $h = g \times t^2 \times 8^{-1}$, where "h" is the height, "g" is the gravity acceleration, and "t" is the flight duration [46]. All jumps were performed at environmental conditions similar to those described for the familiarization procedures and cardiopulmonary exercise testing (~25°C). Each participant performed five trials and was instructed to jump as high as possible. The highest and lowest CMJ heights were excluded, and the arithmetic mean of the three remaining jumps was calculated [23].

**Body core temperature ($T_{CORE}$).** Due to the convenience of the technique and high correlation with rectal probe measurements [47], gastrointestinal temperature, an index of $T_{CORE}$, was monitored throughout the experiment using telemetric capsules (CorTemp® HQ Inc, model HT150002, Palmetto–FL, USA). Because of the temperature variations in the gastrointestinal tract, the participants were instructed to ingest the capsules between 7 h and 8 h before starting the 10 km run. Therefore, all procedures followed the recommendations of Byrne & Lim [48]. Our participants did not report any discomfort caused by ingesting the telemetric capsules.

**Heart rate (HR).** HR was monitored every 2 km during the 10 km run using a Polar H-10 chest strap (Polar Electro Oy, Kempele, Finland), with a sampling frequency of 5 Hz. HR values were recorded by the Polar Beat application (version 2.5.1) and then transmitted to a smartphone, where they were analyzed by the Polar Flow application.

**Perceptual variables.** RPE and thermal sensation were recorded every 2 km during the race. RPE was determined using the 0 to 10-point Borg's Scale [43], whereas thermal sensation was determined by a 7-point scale: cold, cool, slightly cool, neutral, slightly warm, warm, and hot [49, 50]. This scale is suitable for describing a one-dimensional relationship between the physical parameters of indoor environments and subjective thermal sensation.

## Calculated variable

**Heat storage rate.** Heat storage was calculated using the equation proposed by Nielsen [51]: body mass (kg) × specific heat of body tissues (3480 J × °C$^{-1}$ × kg$^{-1}$) x change in $T_{CORE}$ (°C). After that, the heat storage rate was calculated as follows: heat storage × time$^{-1}$ (s) × body surface area$^{-1}$ (m$^2$).

## Statistical analysis

Homogeneity of variance was examined by Levene's test, whereas the data normality was investigated by the Shapiro-Wilk test; no significant effect was reported in either test. Unless otherwise stated, the data were reported as means ± standard deviation (SD). The time to complete the 10 km run and the change in the CMJ height were compared between trials (temperate vs. hot) using paired Student's t-tests. Two-way repeated-measures analyses of variance (ANOVAs) were performed to test differences in physiological and perceptual variables, pacing, and CMJ height between trials and distances (during the race) or between trials and time points (pre- vs. post-race). When significant differences were detected, Tukey's *post hoc* comparisons were performed. A *p*-value < 0.05 was considered to be statistically significant. Statistical analyses were carried out using SigmaPlot software (version 11.0, Systat Software Inc., San Jose—CA, USA).

We calculated Cohen's d effect size as a supplementary analysis to understand findings concerning performance better. This analysis assessed the magnitude of differences between data, and Cohen's d was calculated by subtracting the mean value of one trial from the mean value of the other trial to which it was being compared; the result was then divided by a combined standard deviation of the data. This calculation was further corrected by the correlation between measurements [52]. These analyses were performed using GPower version 3.1 (Universität Düsseldorf, Germany). Effect size values were classified as trivial ($d < 0.2$), small ($d = 0.2–0.6$), medium ($d = 0.6–1.2$), or large ($d \geq 1.2$) [53].

The intraclass correlation coefficient [ICC(3,k)] was calculated for the CMJ height, considering the values obtained before the two 10 km time trials; this calculation was performed in the IBM SPSS software (version 19.0, International Business Machines Corporation, Armonk–NY, USA). We also calculated the standard error of the measurement (SEM) using the following equation [54]: $SEM = SD \times \sqrt{(1 - ICC)}$.

## Results

The participants' characteristics were as follows: age 31.5 ± 6.6 years, body mass 64.7 ± 9.4 kg, height 1.67 ± 0.08 m, body fat 12.3 ± 1.8%, body surface area 1.7 ± 0.1 $m^2$, estimated $VO_{2peak}$ 54.2 ± 2.6 mL·kg$^{-1}$·min$^{-1}$.

The endurance performance was impaired in the heat, as evidenced by the additional 14.1 ± 4.8 min to complete the 10 km time trial at 35˚C than at 25˚C (72.7 ± 10.3 min vs. 58.6 ± 7.6 min; $p < 0.001$; d = 2.93). Indeed, impaired endurance at 35˚C was observed in all 13 participants (Fig 2A) and corresponded to a large effect size. Regarding the average speed to complete 2 km intervals, a significant trial × distance interaction was observed (F = 7.09, $p < 0.001$; Fig 2B). The participants maintained a constant average speed throughout the 10 km at 25˚C but presented a decreased speed from the 4th-6th km interval onwards at 35˚C. As expected, the average speed was always slower at 35˚C than at 25˚C.

Significant main effects of distance were observed for HR (F = 862.02, $p < 0.001$; Fig 3A), $T_{CORE}$ (F = 167.78, $p < 0.001$; Fig 3B), and heat storage rate (F = 23.83, $p < 0.001$; Fig 3C). For example, HR and $T_{CORE}$ were markedly increased during both running exercises, attaining average values above 180 bpm and 39.2˚C, respectively, at the end of the 10 km runs. Neither a significant main effect of trial (F = 0.80, $p = 0.387$) nor a significant trial × distance interaction (F = 1.11, $p = 0.364$) was observed for HR. In contrast, significant interactions were observed for $T_{CORE}$ (F = 2.55, $p = 0.037$) and heat storage rate (F = 4.22, $p = 0.005$). More specifically, despite no differences at the end of running, $T_{CORE}$ was transiently higher at 35˚C than at 25˚C in the 6th and 8th km. The heat storage rate was higher at 35˚C between the 4th and 6th km but lower in the first and final 2 km compared to 25˚C. One hour following the 10 km run,

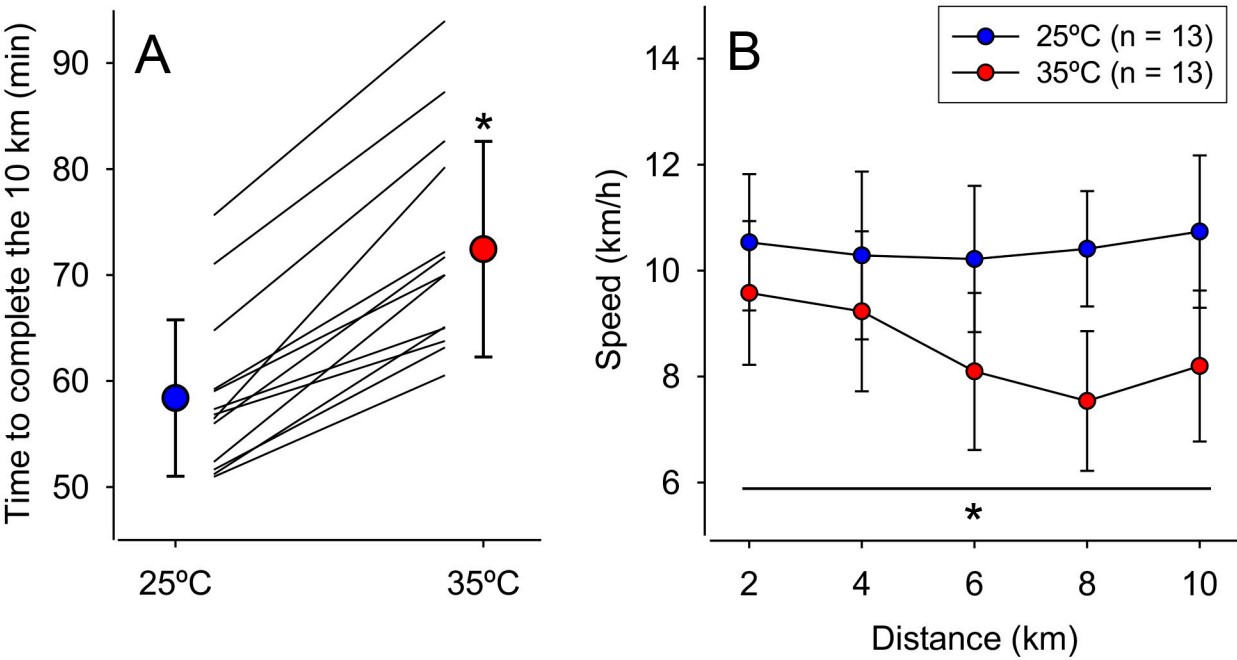

**Fig 2. Endurance performance.** The time to complete the 10 km self-paced run was measured under temperate (25˚C) and hot (35˚C) conditions (panel A). Data are expressed as means ± SD and individual data (*i.e.*, lines); all participants had decreased performance in the heat. Panel B shows the average speed at different distance intervals during the self-paced run in the two environmental conditions. Data are expressed as means ± SD. * indicates a significant difference compared to the control trial at 25˚C, $p < 0.05$.

$T_{CORE}$ was not different between experimental conditions (t = -2.046; $p = 0.063$) and corresponded to 37.49 ± 0.23˚C and 37.59 ± 0.26˚C at 25˚C and 35˚C, respectively.

Regarding the perceptual variables, significant main effects of distance were observed for RPE (F = 105.45, $p < 0.001$; Fig 4A) and thermal sensation (F = 39.40, $p < 0.001$; Fig 4B), which increased markedly during both running exercises. For example, in the hot environment, RPE and thermal sensation corresponded to 9 ± 1 and 7 ± 0, respectively, when participants completed the predetermined distance. For the two variables, significant main effects of trial (RPE: F = 17.64, $p = 0.001$; sensation: F = 148.03, $p < 0.001$), but no significant trials × distance interactions (both $p > 0.430$), were observed.

Before exercise, the CMJ height was not different between the two experimental trials (Fig 5A). CMJ height under baseline conditions corresponded to an ICC(3,k) of 0.984 and an SEM of 0.6 cm. A significant main effect of time point was observed for CMJ height (F = 9.07, $p = 0.001$), with the values being lower at 1 h after (31.2 ± 4.0 cm, d = 1.00, *grouped data from the two trials*) but not immediately after running (32.6 ± 5.1 cm, d = 0.31), compared to before running (33.6 ± 4.4 cm). The reduction observed 1 h after the self-paced exercise corresponded to a medium effect size. Neither a significant main effect of trial (F = 0.33, $p = 0.576$) nor a significant trial × time point interaction (F = 1.36, $p = 0.276$) was observed for the CMJ height. We then compared the reduction in the CMJ height between the two trials immediately after ($p = 0.989$, d = 0.00; Fig 5B) and 1 h after the run ($p = 0.162$, d = 0.40; Fig 5C) and observed no differences; the effect sizes for comparisons between trials were classified as trivial (immediately after) and small (1 h after).

Because one participant had exaggerated reductions in the CMJ height following the run at 35˚C (approximately 11 cm), we repeated the analyses excluding this participant. Again, no

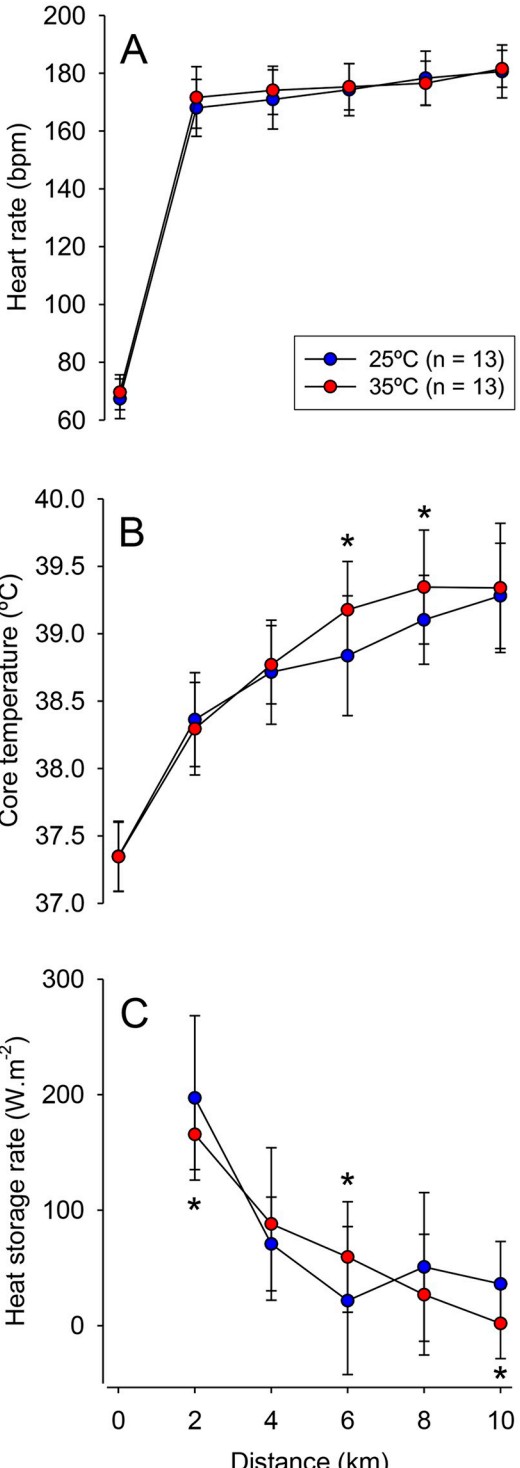

**Fig 3. Physiological responses during the 10 km self-paced run.** The following physiological variables were measured under temperate (25˚C) and hot (35˚C) conditions: heart rate (panel A), core temperature (panel B), and heat storage rate (panel C). Data are expressed as means ± SD. * indicates a significant difference compared to the control trial at 25˚C, $p < 0.05$.

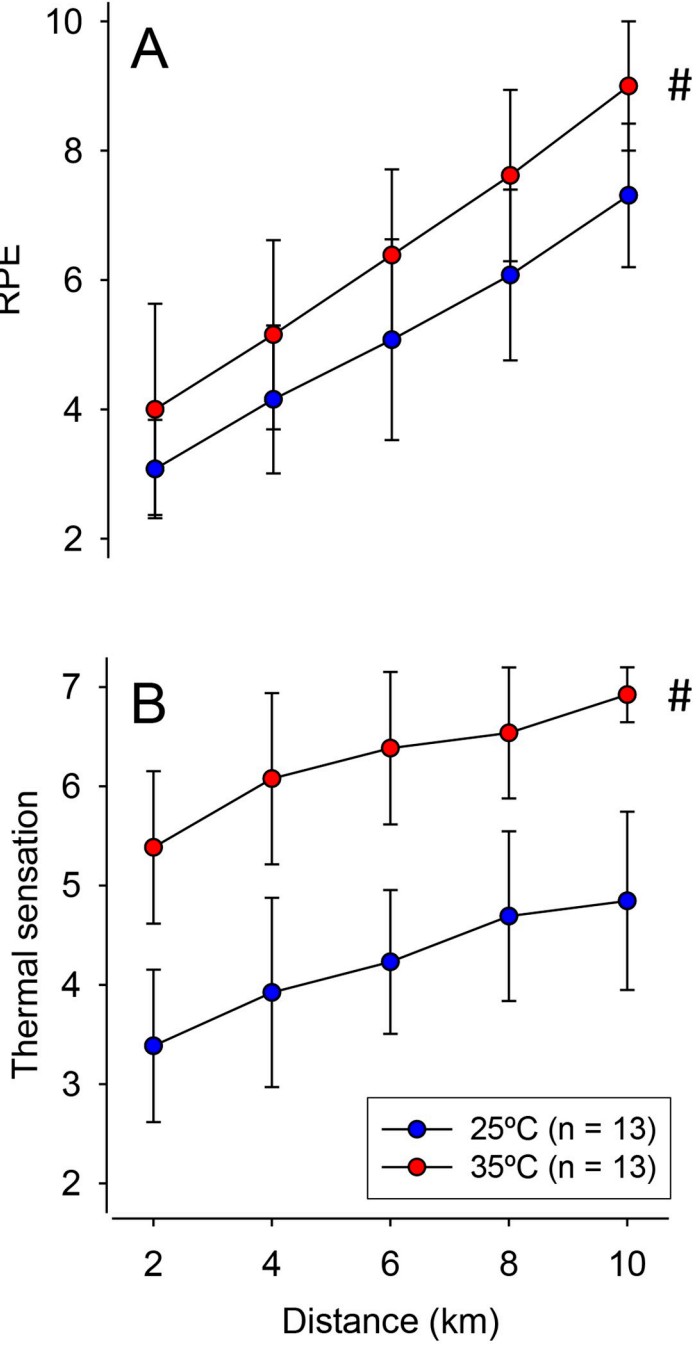

**Fig 4. Perceptual responses during the 10 km self-paced run.** The following perceptual variables were measured under temperate (25˚C) and hot (35˚C) conditions: rating of perceived exertion (RPE; panel A) and thermal sensation (panel B). Data are expressed as means ± SD. # indicates a significant difference compared to the control trial at 25˚C (main effect of ambient temperature), $p < 0.05$.

inter-trial differences were observed in the reduction of the CMJ height immediately after ($p = 0.115$, d = 0.47) and 1 h after the 10 km run ($p = 0.283$, d = 0.29; Fig 5C); these two comparisons were classified as small effect sizes. Therefore, including this participant did not influence the findings regarding CMJ height.

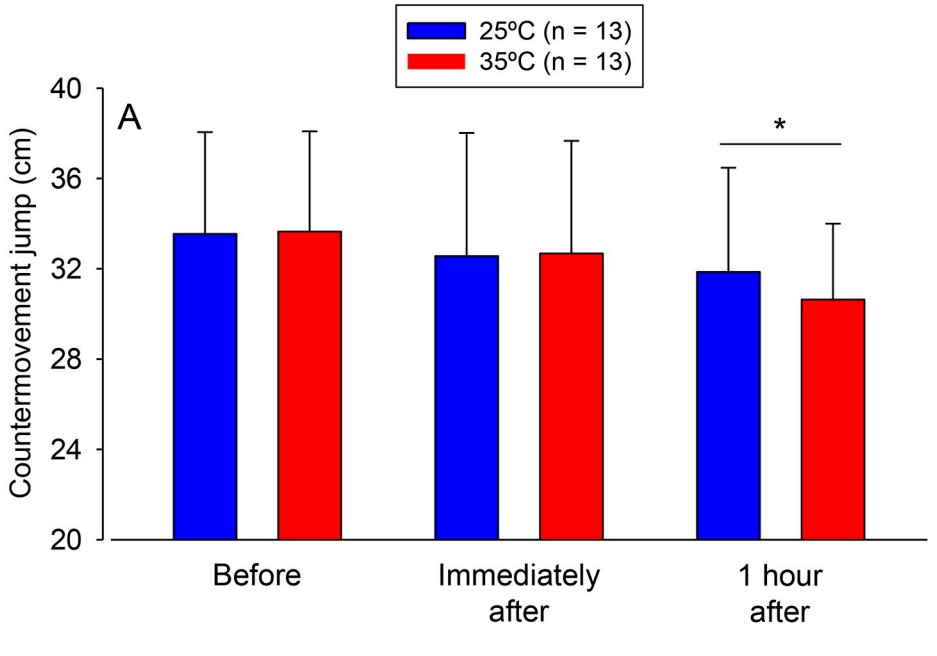

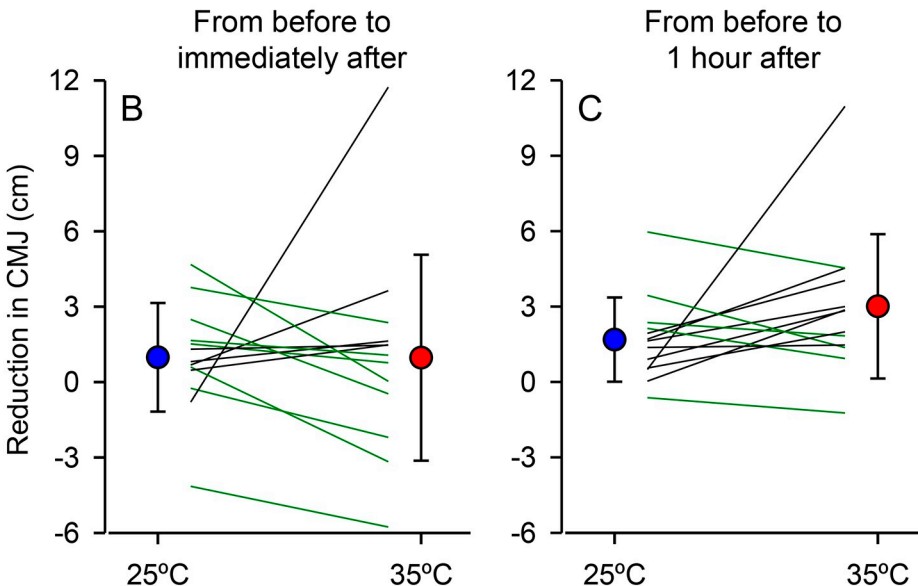

**Fig 5. Lower limb neuromuscular performance before and after running.** The countermovement jump (CMJ) height was measured before, immediately after, and 1 h after the 10 km self-paced run under temperate (25˚C) and hot (35˚C) conditions (panel A). Data are expressed as means ± SD. + indicates a significant difference compared to before running (main effect of time point), $p < 0.05$. Panel B shows the reduction in the CMJ height calculated by subtracting the immediately after from before running, whereas panel C shows the reduction calculated by subtracting the 1 h after from before running. In these two panels, data are expressed as means ± SD and individual data (*i.e.*, lines). The black and green lines indicate, respectively, greater and lower performance reductions after running at 35˚C than at 25˚C.

## Discussion

The objectives of the current study were twofold. First, we assessed the influence of $T_A$ on the time course of physiological and perceptual responses in tropical natives during a 10 km self-

paced run. Second, we investigated the changes in lower limb neuromuscular performance, as determined by CMJ height, following exercise in these two environmental conditions. Our main findings were that, compared to 25˚C, endurance performance was reduced, the perceptual strain was increased, while HR and end-$T_{CORE}$ were unchanged at 35˚C. In addition, CMJ height reduced by 7.0% (2.3 cm on average) 1 h after the run compared to pre-exercise values, and this reduction was not different between the environmental conditions studied. These findings confirm our first hypothesis, except for the end-$T_{CORE}$ that was unaffected by $T_A$ in the current investigation.

As expected, the participants spent additional time (*i.e.*, 14.1 ± 4.8 min) to finish the 10 km run at 35˚C than at 25˚C. This finding corroborates with previous studies on the detrimental effects of hot environments on endurance during self-paced exercises [4, 55, 56], including when individuals ran on motorized treadmills [11, 12, 14, 15]. For example, Marino et al. [14] reported that well-trained athletes struggled to run 8 km as fast as possible at 35˚C. While all participants completed this task at 15˚C, 4 of the 9 participants did not succeed at 35˚C, and even those athletes that could run the 8 km in the heat selected a slower pace.

Our findings contradict the second hypothesis and were somehow surprising because the participants lived in a tropical climate: the average maximum $T_A$ in Teresina was equal to or above 35˚C (*i.e.*, the hot condition in the current study) in 40% of the months in the period comprised between 2013 and 2022 [33]. Therefore, we expected that heat-acclimatized tropical natives would be less sensitive to the performance-impairing effects of environmental heat stress, which was not the case. Our participants reported higher scores of thermal sensation (*i.e.*, greater body sensation of heat) and perceived exertion at 35˚C than at 25˚C throughout the treadmill run. Notably, they usually train early in the morning when $T_A$ and solar radiation are not at their highest levels and are exposed to environments with controlled $T_A$ through air-conditioning in several buildings. Thus, some habits of these tropical natives may limit the full development of a heat acclimatization state. Moreover, previous evidence indicates that adaptive pressure on the thermoregulatory function is more intense during artificial acclimation (*e.g.*, exercise-heat stress) than the pressure associated with residing in a hot climate [57]. In this regard, individuals living in tropical climates also benefit from engaging in a heat acclimation protocol [58], and some adaptations induced by acclimation (*e.g.*, the reduction in heart rate during fixed-intensity walking in a hot environment) did not differ when the protocol was carried out at the end of summer or winter [59].

At 25˚C, the participants maintained a constant pace during the 10 km; *i.e.*, the speed to cover the 2 km intervals ranged from 10.2 to 10.7 km·h$^{-1}$. The selected speed was slower throughout the 10 km at 35˚C than at 25˚C, reproducing previous data obtained in marathons [60] and cycling time trials [61]. At 35˚C, the participants considerably decreased their pace from the 4th-6th km interval until the end of the race. At this interval, the heat storage rate, perceived exertion, and thermal sensation were significantly augmented at 35˚C compared to 25˚C, suggesting that altered physiological and perceptual responses induced by the environmental heat stress contributed to reducing the athletes' pace. Trubee et al. [60] pointed out that non-elite men and women runners, as our participants, may consider implementing a slightly slower initial speed to maintain or increase speed in the latter stages of a race in hotter temperatures to enhance performance. Nevertheless, we did not record a faster speed in the last 2 km, agreeing with the recent findings showing that non-elite runners could not sprint at the end of a 10 km run at 33˚C [62].

$T_{CORE}$ was transiently higher at 35˚C than at 25˚C due to a reduced ability to dissipate the metabolic heat produced while running under hot conditions. Interestingly, when individuals decreased their running speed by ~2.0–2.5 km·h$^{-1}$ at 35˚C, the heat storage rate decreased, and

$T_{CORE}$ became similar between the two environments. This finding disagrees with the observations by Périard et al. [32], who observed a 0.8˚C higher rectal temperature at the end of a 40 km time trial at 35˚C compared to 20˚C. The contradictory results between studies may be explained by the different exercise protocols used (running vs. cycling), $T_A$ of control conditions (25˚C vs. 20˚C), and the more evident reduction in performance we observed compared to Périard's study (24% vs. 8%). The accentuated performance impairment in the current investigation contradicts our expectations since the inhabitants of Teresina lived much closer to the Equator line than the Australian participants of the 2011 study.

At the end of the run, our participants exhibited average $T_{CORE}$ values close to 39.5˚C in both environments. This level of hyperthermia produces significant physiological changes; for example, it is associated with augmented intestinal permeability [63]. A peak $T_{CORE}$ value of 40.3˚C was recorded in an individual, even though the present study consisted of a race simulation, where athletes could select their preferred pace to achieve the best performance while maintaining their physiological responses within safe limits [61]. Still, this peak $T_{CORE}$ is lower than values reported in previous studies investigating thermoregulatory responses during half marathons in hot environments: 40.7˚C [64] and 41.5˚C [65].

When completing the 10 km, the participants were running at 95.5 ± 6.5% (mean ± SD; at 25˚C) and 94.4 ± 4.5% (at 35˚C) of their maximum HR obtained in the incremental test conducted at temperate conditions ($T_A$ = 24˚C and relative humidity = 50%). These data indicate that the individuals exercised near their maximum capacity when they finished the race. Importantly, these similar HR values were attained even with the participants being, on average, 2.5 km·h$^{-1}$ slower during the last 2 km at 35˚C than at 25˚C. Similar observations (*i.e.*, slower speed or lower power output in the heat but similar HR values) were reported by Marino et al. [14], Périard et al. [32], and Maia-Lima et al. [4]. These findings suggest that individuals selected an exercise intensity according to the cardiac strain experienced in two different environmental conditions.

CMJ height decreased 1 h after the 10 km run compared to pre-exercise height. The reduction corresponded, on average, to 2.3 cm, which exceeds by more than three times the SEM, indicating that decreased performance does not result from measurement error. Hyperthermia caused by exercise in the heat reduces voluntary muscle activation. As reviewed by Nybo et al. [66], whole-body hyperthermia is associated with central fatigue as evidenced by direct measures of voluntary activation during sustained isometric contractions or repeated isokinetic contractions, thus leading to lower force production. A previous study reported a lower force development during sustained handgrip contractions in hyperthermic individuals after a cycling exercise in the heat [7]. The authors concluded that the attenuated ability to activate the skeletal muscles in the hand did not depend on whether the muscle group was active during the preceding lower limb cycling exercise [7]. Thus, central fatigue induced by exercise-induced hyperthermia may represent a mechanistic link between prolonged exercise and the reduced CMJ height, even though treadmill running and jumps are substantially different: *e.g.*, metabolic pathways involved in energy supply (aerobic metabolism vs. high-energy phosphates breakdown).

In the current study, the similar $T_{CORE}$ attained at the end and 1 h after the 10 km may help explain the similar reduction in lower limb power in the two $T_A$s investigated. Nevertheless, CMJ height only decreased 1 h after the 10 km, when average $T_{CORE}$ was 37.5˚C, approximately 0.2˚C higher than pre-exercise values. Périard et al. [32] reported that the additional increase in $T_{CORE}$ (using rectal probes) did not exacerbate the loss of force production of the knee extensor following self-paced cycling exercise in the heat. Therefore, the changes in $T_{CORE}$ and the resulting central fatigue (*e.g.*, impaired neural drive; [7]) are not the sole explanation for the reduction in lower limb power, which can also result from

peripheral fatigue (*e.g.*, impaired excitation-contraction coupling). A decline in $Ca^{2+}$ sensitivity, metabolic acidosis, and impaired action potential may occur in contracting muscles during strenuous aerobic exercises, promoting peripheral fatigue and reducing CMJ height [67, 68].

Interestingly, CMJ height was not significantly reduced immediately post-exercise compared to pre-exercise in our experiments. Similarly, CMJ performance was not impaired immediately after simulated tennis matches in cool (21.8°C) and hot environments (36.8°C) [69]. These findings are supported by two recent meta-analyses that indicated no changes in muscle power within 15 min after an endurance exercise session [70] and a delayed reduction in CMJ height following female soccer matches [71]. In the latter meta-analysis, the athletes jumped a lower height at 24 and 48 h following the matches but not immediately after [71]. However, the current data do not agree with an investigation that reported reduced CMJ height immediately, 3 h, and 24 h after a half marathon compared to the pre-race values [72]. Therefore, we cannot rule out that the magnitude and time course of post-exercise changes in CMJ performance depend on the intensity and duration of the previous endurance exercise.

Our data indicate that running 10 km in a hot environment induced similar HR values and neuromuscular fatigue than running the same distance under temperate conditions, although the average speed was considerably lower in the heat. This finding suggests that the higher external load (*i.e.*, the faster speed) at 25°C compensates for the effects of a transiently higher $T_{CORE}$ and more significant perceptual responses at 35°C in inducing neuromuscular fatigue. From an applied perspective, this observation suggests that, at specific training microcycles, exercising under environmental heat stress may be an adequate strategy to maintain the physiological stimulus when a reduction in external load is desired. Notably, heat acclimation was recently proposed to increase training intensity without a subsequent increase in force production in well-trained rowers, thus limiting lumbar spine compressive forces and stress and reducing lesion incidence [73].

This study is not free of limitations. First, lower limb neuromuscular fatigue was assessed only at two points following the 10 km run. Therefore, we must recognize that differences between experimental conditions could exist after the last time point investigated (1 h after the run). Second, among our 13 participants, only two were women. Although we intended to study a similar number of male and female athletes, recruiting women who fulfilled the inclusion criteria was challenging. Finally, our post-exercise neuromuscular fatigue analysis was restricted to CMJ height. Future investigations should also investigate the changes in the force-time profile of these jumps [74] or use a different test protocol (*i.e.*, maximum voluntary contraction or muscular contraction evoked during electrical stimulation) that allows identifying the origin (central or peripheral) of neuromuscular fatigue [7, 75, 76].

In conclusion, hot conditions impair endurance performance, enhance perceptual responses, and transiently changes physiological strain during a self-paced run compared to temperate conditions. Moreover, lower limb neuromuscular fatigue, determined by the reduction in CMJ height, is evident 1 h after the 10 km run. Despite the marked endurance performance impairment induced by exercise-heat stress, post-exercise neuromuscular fatigue is similar between temperate and hot conditions.

## Supporting information

**S1 Data.**
(XLSX)

## Author Contributions

**Conceptualization:** Jefferson F. C. Rodrigues, Júnior, Thiago T. Mendes, Emerson Silami-Garcia, Fabiano T. Amorim, Mário N. O. Sevilio, Jr., Samuel P. Wanner.

**Data curation:** Jefferson F. C. Rodrigues, Júnior, Thiago T. Mendes, Samuel P. Wanner.

**Formal analysis:** Jefferson F. C. Rodrigues, Júnior, Thiago T. Mendes, Fabiano T. Amorim, Samuel P. Wanner.

**Funding acquisition:** Samuel P. Wanner.

**Investigation:** Jefferson F. C. Rodrigues, Júnior, Patrícia F. Gomes.

**Methodology:** Jefferson F. C. Rodrigues, Júnior, Thiago T. Mendes, Patrícia F. Gomes, Samuel P. Wanner.

**Project administration:** Samuel P. Wanner.

**Resources:** Jefferson F. C. Rodrigues, Júnior, Thiago T. Mendes, Fabrício E. Rossi, Samuel P. Wanner.

**Supervision:** Thiago T. Mendes, Mário N. O. Sevilio, Jr., Samuel P. Wanner.

**Validation:** Jefferson F. C. Rodrigues, Júnior, Thiago T. Mendes, Emerson Silami-Garcia, Fabiano T. Amorim, Mário N. O. Sevilio, Jr., Fabrício E. Rossi, Samuel P. Wanner.

**Visualization:** Jefferson F. C. Rodrigues, Júnior, Samuel P. Wanner.

**Writing – original draft:** Jefferson F. C. Rodrigues, Júnior, Samuel P. Wanner.

**Writing – review & editing:** Jefferson F. C. Rodrigues, Júnior, Thiago T. Mendes, Patrícia F. Gomes, Emerson Silami-Garcia, Fabiano T. Amorim, Mário N. O. Sevilio, Jr., Fabrício E. Rossi, Samuel P. Wanner.

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
