## [Decision Letter · Decision Letter 0]

26 Apr 2023

PONE-D-23-01764

Tropical natives exhibit similar lower limb neuromuscular fatigue following a 10 km self-paced running under temperate and hot environments despite reduced running performance in the heat

PLOS ONE

Dear Dr. Wanner,

Thank you for submitting your manuscript to PLOS ONE. After careful consideration, we feel that it has merit but does not fully meet PLOS ONE’s publication criteria as it currently stands. Therefore, we invite you to submit a revised version of the manuscript that addresses the points raised during the review process.

The reviewers have now made recommendations of Major Revision, with most requested changes now related to major concerns about the neuromuscular outcome (countermovement jump) and what is the application of the results. The comments of the reviewer(s) are included at the bottom of this letter.

We look forward to receiving your revised manuscript.

Kind regards,

Leonardo de Sousa Fortes, Ph.D.

Academic Editor

PLOS ONE

Journal Requirements:

Reviewers' comments:

Reviewer's Responses to Questions

**Comments to the Author**

1. Is the manuscript technically sound, and do the data support the conclusions?

Reviewer #1: No

2. Has the statistical analysis been performed appropriately and rigorously? 

Reviewer #1: Yes

3. Have the authors made all data underlying the findings in their manuscript fully available?

Reviewer #1: Yes

4. Is the manuscript presented in an intelligible fashion and written in standard English?

Reviewer #1: Yes

5. Review Comments to the Author

Reviewer #1:

The manuscript entitled “Tropical natives exhibit similar lower limb neuromuscular fatigue following a 10 km self-paced running under temperate and hot environments despite reduced running performance in the heat” aimed to investigate the influence of heat stress on 10 km self-paced running performance and the neuromuscular fatigue impact to this outcome.

First, I would like to thank the opportunity of review a very interesting work. The following comments are just suggestions that aim to help to improve the manuscript.

Despite not particularly new, the manuscript data confirms that running endurance performance is impaired by hot environment (35°C), even for tropical native runners.

In my point of view, the new and noteworthy of the manuscript is the comparison of the time-course of core temperature, psychophysiological variables and running performance (self-paced speed and time to reach 10 km) in each condition.

However, the authors have chosen to emphasize what is, in my opinion, the weakness of the manuscript, the neuromuscular outcome. Although the CMJ performance is highly applicable (and very useful) in the daily training routine, this is a very poor tool to infer any relationship between heat stress X endurance running performance X neuromuscular fatigue.

The characteristics of the test (CMJ) and the exercise (endurance running) simply doesn’t match. Jumps are ballistic exercises which requires in a very short period of time, high level of neuromuscular work (neural drive, muscular machinery), specific demands from the energetic/metabolic systems (energy system and fuel) as well as is highly influenced by technical issues. These requirements are completely different from a continuous run at ~10 km/h. Thus, at the end of the day, that is just a simple attempt to correlate CMJ and running performances.

Also, this test is unable to track the origin of the neuromuscular fatigue (i.e., central or peripheral), and, once the heat stress affects the central neural drive to the active muscles (i.e., voluntary activation), that would be highly desirable. So, in this good manuscript, the CMJ performance is not more than a secondary/complementary data.

I would recommend the following papers that may help this discussion:

Nybo, L., Rasmussen, P., and Sawka, M.N. (2014). Performance in the heat-physiological factors of importance for hyperthermia-induced fatigue. Compr. Physiol. 4,657–689.

(the Central Nervous System role on the neuromuscular fatigue induced by heat stress)

Milioni F, Vieira LH, Barbieri RA, Zagatto AM, Nordsborg NB, Barbieri FA, Dos-Santos JW, Santiago PR, Papoti M. Futsal Match-Related Fatigue Affects Running Performance and Neuromuscular Parameters but Not Finishing Kick Speed or Accuracy. Front Physiol. 2016 Nov 7;7:518.

(heat stress X central fatigue X ballistic exercise (futsal finishing kick)).

Important points that must be addressed:

How the VO2peak was determined? It’s not clear if an ergoespirometer was used. These values must be reported in the results section.

Please, explain better how the average speed was recorded and calculated.

How did the authors keep the room temperature at 25°C and 35°C during the experimental trials? Please, improve the description.

The discussion section must be adjusted to address the conclusive results of the manuscript (time course of the heat stress and running performance at both conditions in tropical native runners).

Minor suggestions

L90 – 93 – Since the author didn’t investigate recovery conditions, I would suggest to remove this sentence.

L130 – 131 – The average temperature reported is regarding a very old time-window (1992 – 2009). Most likely, these values didn’t change significantly, however I would suggest to investigate the records of the closest meteorological center for newer value of the average temperature.

L193 – the authors haven’t reported the body fat percentage in the results. I would suggest to remove.

Results section – please use “variables” instead “parameters”.

6. PLOS authors have the option to publish the peer review history of their article (what does this mean?). If published, this will include your full peer review and any attached files.

Reviewer #1: No

---

## [Author Response · Author response to Decision Letter 0]

29 Jun 2023

PONE-D-23-01764 – Rebuttal letter

Reviewer #1

1. The manuscript entitled “Tropical natives exhibit similar lower limb neuromuscular fatigue following a 10 km self-paced running under temperate and hot environments despite reduced running performance in the heat” aimed to investigate the influence of heat stress on 10 km self-paced running performance and the neuromuscular fatigue impact to this outcome. First, I would like to thank the opportunity of review a very interesting work. The following comments are just suggestions that aim to help to improve the manuscript.

Answer: We thank the reviewer for the relevant comments aimed at improving our study. We took every comment into account and then changed the manuscript accordingly; please see below our responses in a point-by-point fashion.

2. Despite not particularly new, the manuscript data confirms that running endurance performance is impaired by hot environment (35°C), even for tropical native runners.

Answer: The reviewer is correct. Our results confirm considerable data showing that a hot environment impairs endurance performance. As said by the reviewer, it is noteworthy that reduced performance is observed even in tropical native runners often exposed to an environmental temperature of 35�C during their daily activities; this information was further emphasized in the revised manuscript (lines 461 to 464).

3. In my point of view, the new and noteworthy of the manuscript is the comparison of the time-course of core temperature, psychophysiological variables and running performance (self-paced speed and time to reach 10 km) in each condition.

Answer: Again, we thank the reviewer for the fundamental comment. The revised manuscript highlights the time course of physiological and perceptual responses, including the core temperature. To highlight these temporal changes, we edited the following sections of the manuscript, as exemplified as follows: title (lines 1 to 3), abstract (36 to 39), introduction (65 to 91), objectives/hypotheses (124 to 134), discussion (440 to 450), and conclusions (596 to 598). Remarkably, the information order in the abstract, introduction, and discussion sections was switched to highlight the time course of physiological and perceptual responses throughout the revised manuscript.

4. However, the authors have chosen to emphasize what is, in my opinion, the weakness of the manuscript, the neuromuscular outcome. Although the CMJ performance is highly applicable (and very useful) in the daily training routine, this is a very poor tool to infer any relationship between heat stress X endurance running performance X neuromuscular fatigue.

Answer: We agree with the reviewer that other some laboratory-based tests, such as biomechanical analyses including flight and contact time and knee flexion angle at initial contact, would be more specific to running fatigue, thus allowing us to establish more precise relationships between heat stress, endurance performance, and neuromuscular fatigue. However, we intended to use a valid, reliable, practical, low-cost tool that is, therefore, applicable to the daily training routine. The significant reduction in CMJ height observed 1 h after the treadmill run indicates that the proposed tool is meritorious. However, we must recognize that the current manuscript provides a limited contribution to advancing the knowledge about neuromuscular fatigue (its central and peripheral origins) induced by running exercises in the heat. This limitation is now acknowledged at the end of the discussion section (lines 589 to 594).

5. The characteristics of the test (CMJ) and the exercise (endurance running) simply doesn’t match. Jumps are ballistic exercises which requires in a very short period of time, high level of neuromuscular work (neural drive, muscular machinery), specific demands from the energetic/metabolic systems (energy system and fuel) as well as is highly influenced by technical issues. These requirements are completely different from a continuous run at ~10 km/h. Thus, at the end of the day, that is just a simple attempt to correlate CMJ and running performances.

Answer: The reviewer is correct in highlighting that the physiological characteristics of a treadmill run and a CMJ test are different. However, despite the differences mentioned by the reviewer, our data indicate that a 10 km run induces central/peripheral fatigue that can be detected in a maximal jump test. Based on this comment, three sentences were included in the revised manuscripts to acknowledge the differences between a treadmill run and a CMJ test and to emphasize central fatigue as a possible mechanistic link between running exercise and jump performance (lines 532 to 544). Of note, a previous study reported a lower force development during sustained handgrip contractions in hyperthermic individuals after a cycling exercise in the heat (Nybo and Nielsen, 2001; doi: 10.1152/jappl.2001.91.3.1055). The authors concluded that the attenuated ability to activate the skeletal muscles in the hand (handgrip contraction) did not depend on whether the muscle group was active during the preceding lower limb cycling exercise.

6. Also, this test is unable to track the origin of the neuromuscular fatigue (i.e., central or peripheral), and, once the heat stress affects the central neural drive to the active muscles (i.e., voluntary activation), that would be highly desirable. So, in this good manuscript, the CMJ performance is not more than a secondary/complementary data.

Answer: The reviewer is correct. Unfortunately, our method does not allow us to track whether the origin of neuromuscular fatigue is central or peripheral. This limitation was inserted in the discussion of the revised manuscript (lines 589 to 594). In addition, as answered to the third issue raised by the reviewer, several changes were made to the revised manuscript to highlight the time course of changes in physiological and perceptual variables during the self-paced runs in the two ambient temperatures.

7. I would recommend the following papers that may help this discussion:

Nybo, L., Rasmussen, P., and Sawka, M.N. (2014). Performance in the heat-physiological factors of importance for hyperthermia-induced fatigue. Compr. Physiol. 4,657–689. (the Central Nervous System role on the neuromuscular fatigue induced by heat stress)

Milioni F, Vieira LH, Barbieri RA, Zagatto AM, Nordsborg NB, Barbieri FA, Dos-Santos JW, Santiago PR, Papoti M. Futsal Match-Related Fatigue Affects Running Performance and Neuromuscular Parameters but Not Finishing Kick Speed or Accuracy. Front Physiol. 2016 Nov 7;7:518. (heat stress X central fatigue X ballistic exercise (futsal finishing kick)).

Answer: We thank the reviewer for recommending these two relevant manuscripts that helped to improve our discussion. Both of them are cited in our revised manuscript.

Nybo et al. (2014) – lines 532 to 535 – and Milioni et al. (2016) – lines 589 to 594.

8. Important points that must be addressed:

8a. How the VO2peak was determined? It’s not clear if an ergoespirometer was used. These values must be reported in the results section.

Answer: VO2peak was estimated, so the current study did not use a metabolic cart. Because the procedure regarding VO2peak estimation was unclear to the reviewer, we added more information to the methods of the revised manuscript (lines 226 to 231). For example, we wrote that “… resting VO2 was summed to the VO2 corresponding to both the horizontal and vertical components of the last stage completed to estimate VO2peak”. 

As requested by the reviewer, VO2peak and the anthropometric data are now reported in the results section (lines 350 to 352).

8b. Please, explain better how the average speed was recorded and calculated.

Answer: The time elapsed was measured with a stopwatch (precision of 0.01 s), and average speed was calculated as the distance traveled divided by a given time interval. As requested, this information was inserted in the revised manuscript (lines 277 to 279).

8c. How did the authors keep the room temperature at 25°C and 35°C during the experimental trials? Please, improve the description.

Answer: The ambient temperature inside the room where the experiments were conducted was warmed up with the help of an electric heater (model AB1100N, Britânia, Brazil) or cooled down with a split air conditioning system (Springer Midea, Brazil). Whenever required, this split was also used in the dehumidification function. As requested, this information was inserted in the revised manuscript (lines 258 to 261).

 

9. The discussion section must be adjusted to address the conclusive results of the manuscript (time course of the heat stress and running performance at both conditions in tropical native runners).

Answer: Several changes were made to the revised manuscript, including in the discussion section, to highlight the time course of physiological and perceptual responses during the self-paced runs in the two ambient temperatures. Some sentences/paragraphs that were edited are listed as follows: lines 440 to 450, 466 to 468, 491 to 493, 496 to 498, 574 to 577, and 596 to 598. We also highlighted that heat-induced poor endurance performance was observed even in tropical native runners often exposed to an ambient temperature of 35�C during their daily activities (lines 461 to 464).

Minor suggestions

1. L90 – 93 – Since the author didn’t investigate recovery conditions, I would suggest to remove this sentence.

Answer: This sentence was amended (lines 93 to 95) and is now more coherent with the research questions addressed in the current manuscript.

2. L130 – 131 – The average temperature reported is regarding a very old time-window (1992 – 2009). Most likely, these values didn’t change significantly, however I would suggest to investigate the records of the closest meteorological center for newer value of the average temperature.

Answer: We thank the reviewer for the relevant suggestion. The revised manuscript contains updated data regarding environmental conditions in Teresina, Brazil (lines 139 to 143). These data were provided by the Brazilian National Institute of Meteorology (INMET; Instituto Nacional de Meteorologia).

3. L193 – the authors haven’t reported the body fat percentage in the results. I would suggest to remove.

Answer: We ask for apologies for not presenting these data in the original manuscript. The data regarding body fat percentage were included in the results section of the revised manuscript (line 351).

4. Results section – please use “variables” instead “parameters”.

Answer: As requested, we replaced “parameters” with “variables” throughout the results section.

---

## [Decision Letter · Decision Letter 1]

2 Aug 2023

Reduced running performance and greater perceived exertion, but similar post-exercise neuromuscular fatigue in tropical natives subjected to a 10 km self-paced run in a hot compared to a temperate environment

PONE-D-23-01764R1

Dear Dr. Samuel Penna Wanner,

We’re pleased to inform you that your manuscript has been judged scientifically suitable for publication and will be formally accepted for publication once it meets all outstanding technical requirements.

Kind regards,

Leonardo de Sousa Fortes, Ph.D.

Academic Editor

PLOS ONE

Reviewers' comments:

Reviewer's Responses to Questions

**Comments to the Author**

1. If the authors have adequately addressed your comments raised in a previous round of review and you feel that this manuscript is now acceptable for publication, you may indicate that here to bypass the “Comments to the Author” section, enter your conflict of interest statement in the “Confidential to Editor” section, and submit your "Accept" recommendation.

Reviewer #1: All comments have been addressed

2. Is the manuscript technically sound, and do the data support the conclusions?

Reviewer #1: Yes

3. Has the statistical analysis been performed appropriately and rigorously? 

Reviewer #1: Yes

4. Have the authors made all data underlying the findings in their manuscript fully available?

Reviewer #1: Yes

5. Is the manuscript presented in an intelligible fashion and written in standard English?

Reviewer #1: Yes

6. Review Comments to the Author

Reviewer #1: The authors have properly addressed all my suggestions. I would like to congratulate for the great work!

7. PLOS authors have the option to publish the peer review history of their article (what does this mean?). If published, this will include your full peer review and any attached files.

Reviewer #1: **Yes: **Fabio Milioni

---

## [Editor Report · Acceptance letter]

8 Aug 2023

PONE-D-23-01764R1 

Reduced running performance and greater perceived exertion, but similar post-exercise neuromuscular fatigue in tropical natives subjected to a 10 km self-paced run in a hot compared to a temperate environment 

Dear Dr. Wanner:

I'm pleased to inform you that your manuscript has been deemed suitable for publication in PLOS ONE. Congratulations! Your manuscript is now with our production department. 

Kind regards, 

on behalf of

Dr. Leonardo de Sousa Fortes 

Academic Editor

PLOS ONE